# Clinical Changes of Respiratory Parameters in Institutionalized Older Adults after a Physiotherapy Program Combining Respiratory and Musculoskeletal Exercises

**DOI:** 10.3390/healthcare10091680

**Published:** 2022-09-02

**Authors:** Anna Arnal-Gómez, Manuel Saavedra-Hernández, Antonio Martinez-Millana, Gemma V. Espí-López

**Affiliations:** 1Department of Physiotherapy, Faculty of Physiotherapy, University of Valencia, Gascó Oliag Street, 5, 46010 Valencia, Spain; 2Physiotherapy in Motion, Multi-Speciality Research Group (PTinMOTION), University of Valencia, 46010 Valencia, Spain; 3Department of Medicine, Nursing and Physical Therapy, University of Almeria, Carretera Sacramento, s/n, 04120 Almería, Spain; 4ITACA, Universitat Politècnica de València, Edificio 8G-Camino de Vera, s/n, 46022 Valencia, Spain; 5Exercise Intervention for Health (EXINH), University of Valencia, 46010 Valencia, Spain

**Keywords:** breathing exercises, respiratory function tests, older adults, residential facilities

## Abstract

Nowadays pulmonary diseases are an increasingly important cause of morbidity and mortality. Diaphragmatic breathing is a controlled-breathing technique that aims to optimize thoracoabdominal movements. The aim of this study was to apply a respiratory and musculoskeletal physiotherapy program in institutionalized older adults and to assess the effects on their pulmonary function tests and oxygen saturation. A randomized double-blind clinical trial was conducted with thirty institutionalized older adults, randomly assigned to a control group (CG), who conducted musculoskeletal exercises; or an experimental group (EG) who, in addition, carried out diaphragmatic breathing, administered for eight weeks, three times/week. Outcomes were pulmonary function variables (forced vital capacity, FVC; forced expired volume at 1 s, FEV1; the FEV1/FVC ratio) and oxygen saturation (SpO_2_) before and after treatment. Normality of the distributions was tested with Saphiro-Wilk and the pre-post improvement was assessed with a two-sample Mann-Whitney test. Significance level was corrected for multiple comparisons using Benjamini-Hochberg correction (*p* < 0.04). There was a clinically significant improvement of FVC and FEV1 for the EG. Moreover, the EG showed a statistically significant increase of SpO_2_ (*p* = 0.028) after treatment when compared to CG. A physiotherapy program combining breathing and musculoskeletal exercises, improved respiratory parameters in institutionalized older adults.

## 1. Introduction

Worldwide, the number of older adults is expected to more than double by 2050 and triple by 2100, mostly in developing countries [1]. Aging involves a gradual reduction in the capacity to adapt and an increased vulnerability to health issues [2], particularly affecting the respiratory system. In fact, aging implies anatomical and functional changes in the respiratory system and, therefore, lung ventilation becomes more superficial and the capacity to adapt to exercise decreases in as much as 30 percent [3]. Moreover, according to the World Health Organization, respiratory diseases such as trachea, bronchus or lung cancer, and chronic obstructive pulmonary disease (COPD) or tuberculosis are found among the leading causes of death [4]. In addition, one of the most significant pandemics of the modern era has recently emerged, namely, Coronavirus Disease 2019 (COVID-19) due to a novel coronavirus [5], producing a severe acute respiratory syndrome [6], which poses a higher risk for older adults [7].

Therefore, pulmonary diseases are an increasingly important cause of morbidity and mortality and have become a major public health problem and a challenge for physical therapists, both due to the large number of people affected and to the potential severity [8]. Thus, scientific societies highlight the importance of prevention of these diseases [9] in order to protect respiratory health, particularly in high risk populations, before symptoms appear [8].

In the last decade, research on preventive respiratory physical therapy has been conducted, although mostly focused on respiratory muscle training [10,11,12]. In relation to breathing strategies, previous research has also shown evidence for the effectiveness of controlled breathing on dyspnea, mainly for patients with COPD [13]. Studies have shown that people with COPD who undergo breathing training, such as yoga breathing [14], pursed-lips [15], or computer-aided feedback [16], can adopt a slower, deeper breathing pattern thus improving efficiency [17].

Studies on breathing exercises often investigate chronic respiratory diseases and rarely involve older population without pathological conditions. Diaphragmatic breathing is a controlled-breathing technique that aims to optimize thoracoabdominal movements whereby the patient is told to move the abdominal wall predominantly during inhalation and to reduce upper rib cage motion [13]. This may prevent thoracic deformation and decrease the energy cost of breathing, contribution of rib cage muscles and dyspnea [18]. Furthermore, musculoskeletal exercises in older people not only promote movement and independence but may also improve certain conditions that underlie disability in older adults, including cardiovascular or respiratory diseases [19,20].

However, to our knowledge, no previous study has addressed the efficacy of combining diaphragmatic breathing with musculoskeletal exercises in institutionalized healthy older adults. Thus, a simple and low-cost physiotherapy program that may improve certain respiratory parameters could be of great value.

The aim of the present study was to apply an easy and reproducible respiratory and musculoskeletal physiotherapy program in institutionalized older adults and to assess the effects on respiratory parameters such as pulmonary function tests and oxygen saturation. Our hypothesis was that a physiotherapy program that combines diaphragmatic breathing with musculoskeletal exercises could improve the respiratory parameters in institutionalized older adults.

## 2. Materials and Methods

### 2.1. Participants

A sample of 40 institutionalized older adults was recruited from the residential facility “Jardines del Palau” in the city of Valencia. The inclusion criteria were: men and women aged over 60; with no previous clinical history of respiratory health issues; familiarized with musculoskeletal exercises; and able to provide informed consent and follow instructions (Mini Mental State Examination ≤ 25 points) [21]. The exclusion criteria were: having respiratory conditions such as COPD, asthma, bronchitis, pneumonia, or lung cancer; smoking; musculoskeletal injuries or dysfunctions that did not allow to perform exercises; and severe cognitive impairment that affected speech or understanding. All data were collected at the University of Valencia.

### 2.2. Study Design

A randomized double-blind clinical trial was carried out from January to March 2019. Participants were randomly allocated to an experimental group (EG), which followed a physiotherapy program that combined diaphragmatic breathing with musculoskeletal exercises, or a control group (CG) whose participants conducted musculoskeletal exercises. Both groups carried out three 40-min sessions per week of exercises during eight weeks, while the EG additionally conducted three 20-min sessions per week of diaphragmatic breathing. Two assessments were carried out: at baseline and after the eight-week intervention. Treatments were applied by a respiratory physiotherapist also specialized in older adults, with more than 10 years of clinical experience. Participants provided informed consent following an explanation of the study aims and procedures before entering the study. The current clinical trial was conducted following the CONSORT extension for pragmatic clinical trials [22]; it complies with the Declaration of Helsinki and was approved by the University of Valencia Ethics Committee (H1539944398481). Trial registered at www.clinicaltrials.gov (NCT03782779) (accessed on 20 May 2020).

### 2.3. Randomization and Blinding

Participants who fulfilled the inclusion criteria were assessed by a single researcher, a physiotherapist with more than 15 years of clinical experience and specialized in assessments in clinical trials, who was not aware of the interventions. They were then randomly assigned to EG or CG by an assistant who did not participate in the trial. Sequentially numbered envelopes were prepared with random assignment and the neutral assistant retained the randomized intervention list throughout the entire duration of the study. Coding, analysis, and interpretation of results were conducted by an external assistant.

### 2.4. Interventions

All interventions were performed in a common area of the residential facility, with enough space for all participants to be seated. Conditions of the room were at a constant of 22 ± 2 °C and humidity of 50 ± 5%. The program was performed at 10:00 a.m. on Mondays, Wednesdays, and Fridays. Participants were asked to perform all study intervention sessions. On Tuesdays and Thursdays, they performed their usual routine activities with other professionals of the residential facility: cognitive activities, walks around the facility, medical appointments, etc. Nutrition remained the same throughout the week and was controlled by the physicians of the residential facility according to participants conditions or needs. The same applied to medications, which were adjusted, if needed, by the physicians of the facility.

Musculoskeletal exercises: Participants were seated in a circle so that they could see each other and the physiotherapist and followed a protocol of active range of motion exercises based on clinical guidelines [23] and previous studies [24]. Sessions took place at the institution of the participants. Musculoskeletal exercises are commonly practiced in residential facilities, and all participants had previously practiced them to some extent with their physiotherapist. Specifically in our study, each session consisted of a 10-min warm-up; 10 min of upper limb exercises involving opening and closing the hand, wrist and elbow flexion-extension, and shoulder flexion-extension and abduction-adduction; 10 min of lower limb exercises including ankle and knee flexion-extension, and hip flexion-extension and abduction-adduction (Appendix A). The sessions ended with a 10-min cool-down by stretching lower and upper limb muscles. Exercises were shown and explained using simple and short sentences. Each exercise was performed for one minute, with each participant doing as many repetitions as he/she could, followed by a one-minute rest period; after four weeks the time was increased to two minutes each exercise [25].

Diaphragmatic breathing: EG in addition to musculoskeletal exercises performed diaphragmatic breathing in a sitting position conducted before or during the exercises when required. It consisted of six steps [8,26] applied gradually (Table 1). Step one: Comfortable body position and breathing awareness, participants were asked to take deep breaths and observe each other in order to detect thoracic movements associated with their breathing; they were also instructed to relax their shoulders and upper chest, highlighting the role of nasal inhalation; Step two: Learning and applying diaphragmatic breathing, by inhaling through the nose and applying diaphragmatic movements so that the diaphragm contracts with greater amplitude, visualizing how this inspiratory descent moves the abdominal area outwards, allowing the respiratory frequency to become slower, followed by oral passive exhalation and awareness of the inward movement of the abdominal region; Step three: Diaphragmatic breathing for coughing, inhaling through the nose, using diaphragm movement as much as they could and then performing a sudden contraction of the abdominal muscles while coughing; Step four: Diaphragmatic breathing together with musculoskeletal exercises (Figure 1); Step five: Diaphragmatic breathing during activities of daily living (ADL), attempting to internalize the new pattern throughout their daily routine. Coordinating inhalation and exhalation breathing cycles during certain ADL were applied to conserve energy. With more intense activities, such as getting dressed, brushing hair, or tying shoelaces, patients were taught to firstly inhale and then exert the effort while exhaling, as many times as needed to complete the task [27]; Step six: Consolidation of the breathing pattern.

### 2.5. Outcomes

Firstly, anthropometric, and clinical variables were registered. Anthropometric variables were: (1) Age and sex; (2) body weight (kg), assessed using a Tanita BC 601 model weighing device (TANITA Ltd., Amsterdam, Netherlands); (3) barefoot standing height (cm), measured with a stadiometer SECA 213 (Seca Ltd. Hamburg, Germany); (4) body mass index, calculated based on the weight (kg) parameters divided by height squared (m^2^). Clinical variables were: (1) Number of days per week that participants did exercise in the last 10 years; (2) number of medications taken daily and regularly; (3) hospital stays in the last five years (recorded as “yes” or “no”).

Secondly, the following assessments were conducted:

Pulmonary function tests (PFTs): a spirometry using a bidirectional digital turbine flowmeter desktop spirometer (Pony FX, COSMED Srl, Rome, Italy) was performed to obtain forced vital capacity (FVC), forced expired volume at one second (FEV1) and the FEV1/FVC ratio (%) by forced expiratory flow maneuver. Patients’ age, height, weight, sex, and ethnicity were introduced in the spirometer, so it predicted lung size and the reference value. Participants were then seated on a chair with back support, wore a nose clip and received instructions to prevent air leaks from around the flange of the mouthpiece. The test was performed according to the American Thoracic Society standards [28] and the highest value of FEV1 and FVC was selected from three repeated measurements in not more than eight tests. For FVC and FEV1, the predicted percentage was registered and 80% or over of the reference value was considered as an optimal value [8]; for the FEV1/FVC ratio, a decreased percentage allowed defining the concept of “obstruction” [18]. A minimal clinically important difference (MCID) is considered ≥11% for FVC and ≥12% for FEV1 [29]. Among the different lung volume measurement methods, spirometry is the most common pulmonary function test [18,28], with an intraclass correlation coefficient of 0.890 (95%CI 0.837–0.926) in COPD patients [30].

Pulse oximetry: Oxygen saturation (SpO_2_) was assessed using a finger pulse oximeter, SmartOx (WEINMANN, Medical Technology, Hamburg, Germany). Participants remained seated and rested, and the pulse oximeter was placed on the index finger, clean and free of nail polish. Several seconds were allowed for the pulse oximeter to detect the pulse and calculate the oxygen saturation [4]. Oxygen saturation levels around 96% to 100% are considered normal at sea level [31]. The significant clinical change was calculated and considered 1.5% in SpO_2_. Oxygen saturation is frequently used in the evaluation and monitoring of patients with respiratory conditions, it has shown to be a reliable method for detecting hypoxemia and has been reported accurate in reflecting one-point measurements of SpO_2_ [32].

### 2.6. Statistical Analysis

The data are described using the mean and standard deviation (95% CI) for continuous variables and the relative distribution for categorical variables. Due to the reduced sample size, normality of the assessment variables was analyzed with the Shapiro-Wilk test. Statistical comparisons between the two groups were analyzed using the two-sample T-Student test for unpaired samples for variables with a normal distribution and the two-sample comparison procedure consisting of a non-parametric Mann-Whitney test to compare intragroup improvement (pre-post intervention). All statistical analyses were carried out using the Statistical Package for Social Sciences (SPSS 23, SPSS Inc., Chicago, IL, USA) for Windows. The tests assumed a significant improvement of 13% in PFTs and 1.5% in SpO_2_, leading to a statistical power of 80% for the considered sample. Multiple comparisons significance level was calculated with the Benjamini-Hochberg correction with a false discovery rate of 25% (*p* < 0.04). Moreover, the effect size (Cohen’s d) [33] was calculated for comparisons where statistically significant differences were obtained. According to the Cohen method, the magnitude of the effect could be considered small (0.20–0.49), moderate (0.50–0.79), or large (>0.8).

We carried out a non-probability sampling, for the convenience of the investigation. Sample size was calculated using the Granmo calculator v.7.12, based on the analysis of two independent means, and estimating an alpha risk of 5% (0.05), a beta risk of 10% (0.10), in a one-tailed test, a typical deviation of 11.5% (0.115), a minimum expected difference to detect of 13% (0.13) which is based on the minimum clinically important differences in the FEV1/FVC ratio [29], and a follow-up loss rate of 8%, for which 15 subjects are required in each group, assuming that there are two groups. Ultimately, we included 40 patients divided into two groups, to account for the possible loss at follow-up and dropouts.

## 3. Results

From the forty prospective participants, thirty-two individuals were eligible, agreed to participate, and were randomly assigned to a group. Figure 2 shows the recruitment process and dropouts.

Finally, thirty participants performed all the sessions and completed the study (EG, *n* = 15; CG, *n* = 15). No participants experienced adverse effects. The mean age was 77.93 ± 9.07 years (minimum: 60, maximum: 91), with 70% of the sample being women (*n* = 21). No participants were hospitalized during the treatment, nor changes in medication which could influence the program were reported to researchers. Demographic and clinical characteristics of the participants are depicted in Table 2.

A general improvement in the mean values of all PFTs and pulse oximetry variables was observed in the EG in relation to the CG after the implementation of the study. Table 3 shows the statistical values and the significant differences and the minimal clinical changes of the variables between and within groups. Comparison between the EG and the CG after treatment showed no significant differences for FVC (*p* = 0.329, z = 0.975), FEV1 (*p* = 0.7243, z = 0.353) and the FEV1/FVC ratio (*p* = 0.967, z = 0.041). FVC and FEV1 values exhibited a clinically significant improvement for the EG (≥11% for FVC and ≥12% for FEV1).

Moreover, the FVC of participants hospitalized in the last five years improved in the EG, however, not in the CG (*p* = 0.01, t = −2.74). In this same line, the results show that EG participants who had done mild exercise in the last ten years (1–2 days/per week) improved their FVC and FEV1 values.

When comparing SpO*_2_* between groups this increased significantly in the EG after the eight-week treatment with a large effect size (*p* = 0.028; z = 2.19), and this increase was also above the clinically significant change (>1.5%; d = 0.8). No SpO*_2_* within group changes of the EG and the CG showed statistical or clinical significance in the post-assessment. Figure 3 summarizes the distribution of post-intervention data for the observed values of FVC, FEV1, and SpO_2_.

## 4. Discussion

According to our results, and considering the intrinsic limitations, this study shows that a physiotherapy program that combines diaphragmatic breathing with musculoskeletal exercises may improve respiratory parameters in institutionalized older adults. Particularly, there was a minimal clinical important change for FVC and FEV1 in the EG. In relation to SpO**_2,_** there were statistically and minimal clinical differences between groups. In relation to our participants, most of them were women and had three or more medications per day, however, they had not been hospitalized and had exercised regularly in the last years.

With regard to pulmonary function variables, the results of the present study showed a clinically significant improvement in the EG for both FVC and FEV1 (>12%) [29]. These results, together with the consideration that the diaphragm is the principal generator of tidal volume in healthy people at rest [34], may explain that the diaphragmatic breathing pattern applied may have led to the clinical improvement of the spirometry variables in the EG. Diaphragmatic breathing focuses on activating the diaphragm during inhalation and, at the same time, minimizing the role of accessory muscles [35], thus it has the potential to improve tidal volume [26]. Considering that aging is characterized by osteoarticular stiffness of the rib cage, dehydration of intervertebral discs and progressive ossification of chondrocostal and costovertebral joints, this explains why older people suffer from chest stiffness and marked dorsal kyphosis [3]. PFTs carried out in our study likewise indicated a trend towards restrictive characteristics. In this regard, initially both groups exhibited a moderately abnormal spirometry (FEV1%: 60–69) and values only improved in the EG to a mild type of abnormality after the intervention (FEV1% > 70) [29]. Overall, the increase of FVC and FEV1, plus the reduction of the FEV1/FVC ratio, indicate suspicion of a slight improvement in the initial restrictive characteristics of the EG participants. Therefore, the combination of diaphragmatic breathing and musculoskeletal exercises involving the upper and lower limbs may have improved the restrictive situation. This is in line with previous studies, that have pointed out that restricted thoracic expansion [36], reduced respiratory muscle strength [37], and impaired physical performance [38] may reduce lung volume, particularly in older adults.

Regarding SpO_2_, the result of the present study not only showed a statistically significant difference between groups but also a large effect size. Considering that pulse oximetry is accepted as indicative of respiratory failure (arterial SpO_2_ < 90%) [39], these results are promising since the technique is commonly available, easy to measure and one of the major parameters used in clinical practice [20]. Previous research has stated that ventilatory response to lower oxygen tension is impaired in older adults [40] due to age-associated physiological changes such as decline in efferent neural output to respiratory muscles, and is also related to reduction of inspiratory airflow [41]. Therefore, the diaphragmatic breathing applied in the EG, may have helped to lower the breathing effort and consequently enhance the oxygen cost of breathing, thus improving the SpO_2_. This is in line with other studies that have reported that slow and deep breathing improves breathing efficiency and oxygen saturation at rest [13].

Another important aspect to consider is that there is a high prevalence of disability associated with aging [38], partially due to the loss of skeletal muscle mass which may also affect respiratory muscles [42,43]. Considering that institutionalized older adults seem more prone to these risk factors [11], they may be unable to undergo whole-body exercise training (e.g., walking) because of comorbid conditions. Therefore, the gentle treatment proposed in this study is a useful alternative to prevent the clinical deterioration in this population with disabilities who are susceptible to disease. In this sense, our results must be taken with caution since in our sample there were participants between 60 to 91 years old, which is an age range were there may be different levels of impairments. Therefore, analyzing the variables depending on age ranges could be considered in future studies with a larger sample. On the other hand, our sample consisted mostly of women, which could be considered as a bias, and should also be addressed if this study is replicated, trying to have a homogeneous sample in relation to sex. Considering that aging is associated with physiological changes and loss of respiratory functional mechanisms, plus the current pandemic situation to which this population can be more vulnerable, these results reinforce the importance of respiratory system-related maintenance and prevention approaches in residential facilities.

In terms of study limitations, the main one is related to the internal validity of the analysis due to the limited sample size. Nevertheless, the moderate to large effect size and the results of the significant value indicated credible statistical analysis despite such a small sample size [44]. Besides, the baseline characteristics of the study sample must be considered. In this sense, the experimental group showed a slightly higher FVC and SpO**_2_** in the pre-intervention, which may have affected enhanced capacity of this group to benefit with the specific program. Moreover, the sample was mostly women, although the proportion is characteristically related to the aged population in Spain. In addition, some more information regarding specific medication participants were having or certain conditions such as hypertension, could have been registered to characterize the sample. However, since participants were institutionalized, these aspects were controlled by their physicians. Thus, our results must be taken with caution and further studies with larger sample sizes and greater equality between sexes would be recommended to reduce possible internal validity threats and confirm our promising results. Future research could also include functional outcome measures and their association with pulmonary function in order to increase clinical relevance.

One of the strengths of this study is that it offers residential facility health personnel an affordable and reproducible intervention and assessment in order to prevent and improve respiratory aging characteristics and reduce the vulnerability of older adults.

## 5. Conclusions

The results of the present study suggest that a physiotherapy program, based on gentle respiratory exercises of diaphragmatic breathing added to a basic program of musculoskeletal exercises, may improve respiratory parameters such as FVC, FEV1, and SpO_2_ in institutionalized older adults. This program offers a useful alternative for healthcare professionals treating older adults in order to implement strategies to improve their health.

## Figures and Tables

**Figure 1 healthcare-10-01680-f001:**
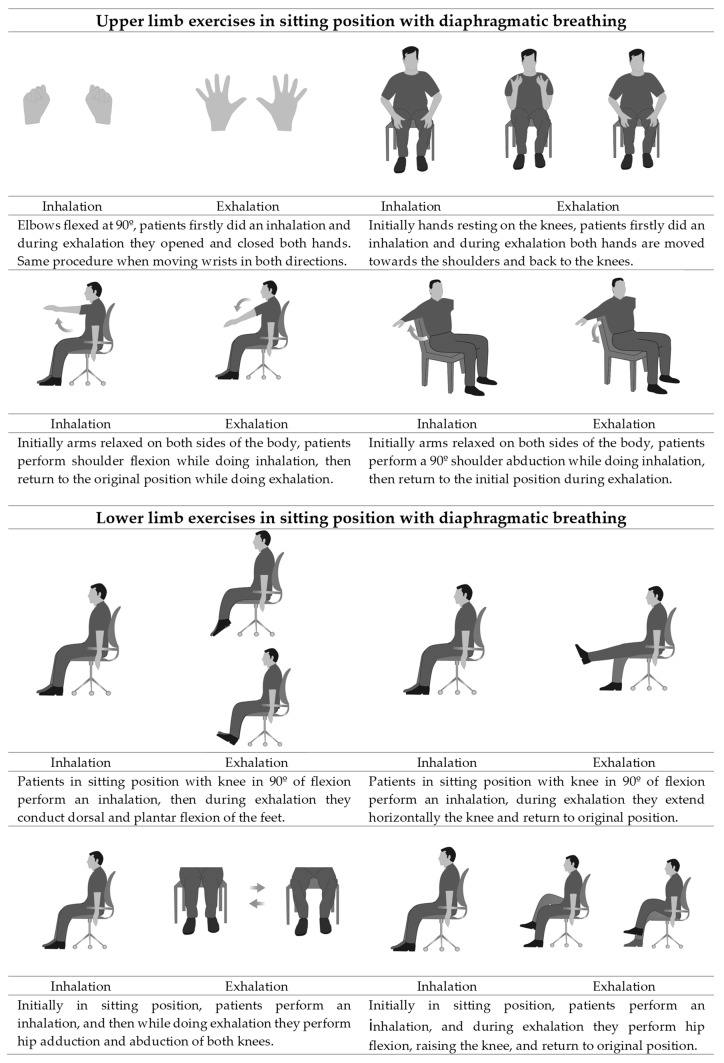
Diaphragmatic breathing combined with musculoskeletal exercises.

**Figure 2 healthcare-10-01680-f002:**
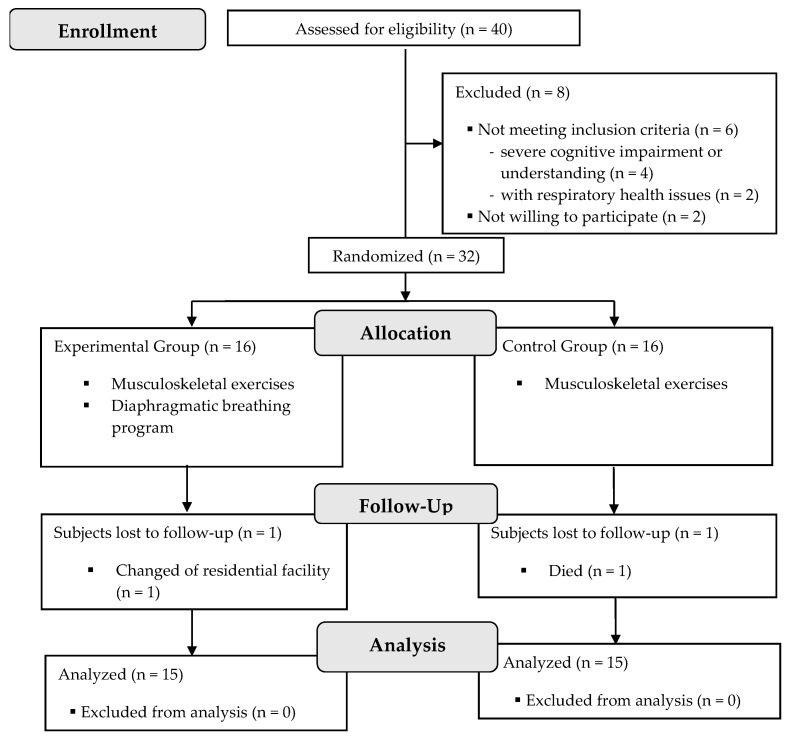
Flowchart according to the CONSORT statement for the report of randomized trials.

**Figure 3 healthcare-10-01680-f003:**
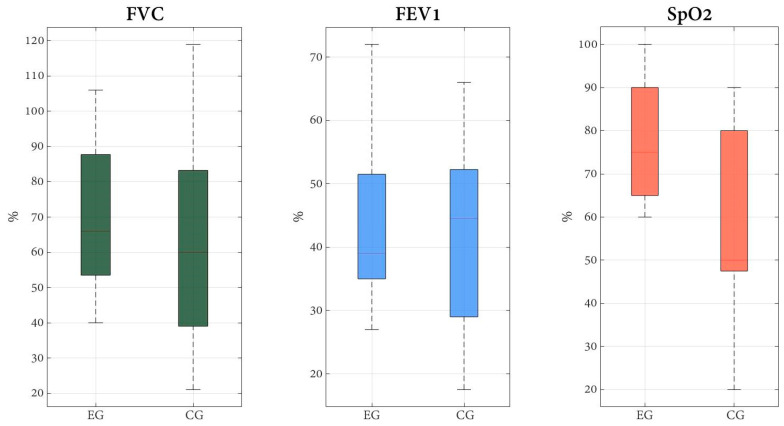
Post intervention comparison of the outcome variables FVC, FEV1, and SpO_2_ for the Experimental Group (EG) and the Control Group (CG).

**Table 1 healthcare-10-01680-t001:** Organization of weeks, sessions, and steps of the diaphragmatic breathing for the experimental group.

Week	Session	Step
1	1	1st Comfortable body position and breathing awareness: with appropriate position of the pelvis, neck, eyes, and upper and lower extremities, participants took deep breaths and observed each other’s breathing chest and/or abdominal movements.
	2	
2	3	2nd Learning and applying diaphragmatic breathing: participants were instructed to apply nasal inhalation and oral exhalation for promoting a more even inhalation, and patency of small airways during exhalation, respectively. Participants hands were placed on the abdomen (level of the umbilicus) and upper chest (level of the manubrium) for tactile stimulation of movements, and for visual stimulation, participants were instructed to observe increased motion of hand over the abdomen and decreased motion of hand over the upper chest.
	4	
	5	3rd Diaphragmatic breathing for coughing: participants were instructed to inhale through the nose with the abdominal-diaphragmatic breathing pattern and then incorporate an abdominal contraction when coughing.
	6	
3	7	4th Diaphragmatic breathing during exercises: for upper limb exercises patients alternated one arm and the other, firstly doing an inhalation while rising the arms and then lowered it in exhalation; for lower limb exercises patients firstly did an inhalation and then they applied the strenuous movement of the exercise while doing exhalation; physiotherapist provided audible stimulation by inhaling loudly with the inspiratory maneuver of the patient and exhaling loudly with the expiratory maneuver of the patient.
	8	
	9	
4	10	
	11	5th Diaphragmatic breathing during activities of daily living: participants were instructed to apply the new pattern gradually in their activities, by performing inhalation when pulling movements and elevation of the arms and exhalation with the strenuous parts of an activity (tying shoelaces) and with pushing movements and lowering of the arms.
	12	
5–8	13–24	6th Consolidation and resolution of doubts: In order to consolidate the breathing technique, during the last four weeks, the diaphragmatic breathing was applied during upper and lower limb exercises and in activities of daily living.

**Table 2 healthcare-10-01680-t002:** Baseline comparisons of demographic and clinical data.

	CG (*N* = 15)	EG (*N* = 15)	P
Anthropometric variables			
Age, mean (SD)	75.8 (9.6)	80.06 (8.2)	0.180
Sex, frequency (%)			0.690
Male	5 (33.3)	4 (26.7)	
Female	10 (66.7)	11 (73.3)	
Weight (kg), mean (SD)	59.5 (11.3)	61.0 (8.8)	1
Height (m), mean (SD)	1.6 (0.1)	1.6 (0.1)	0.441
BMI (kg/m^2^), mean (SD)	24.5 (3.8)	24.6 (2.9)	0.775
Social and clinical variables			
Marital status, frequency (%)			
Unmarried	2 (13.3)	3 (20.0)	0.856
Married	5 (33.3)	4 (26.7)	
Widower	8 (53.3)	8 (53.3)	
Educational level, frequency (%)			
Non-complete primary level	3 (20.0)	4 (26.7)	0.828
Primary level	7 (46.7)	4 (26.7)	
Secondary level	2 (13.3)	3 (20.0)	
Professional training	2 (13.3)	2 (13.3)	
University degree	1 (6.7)	1 (6.7)	
Number of days per week of exercise, in the last 10 years, frequency (%)			
0	4 (26.7)	5 (33.3)	0.649
1–2	4 (26.7)	3 (20.0)	
3 or more	5 (33.3)	7 (46.7)	
Medication, frequency (%)			
0	1 (6.7)	1 (6.7)	0.746
1–2	3 (20.0)	5 (33.3)	
≥3	11 (73.3)	9 (60.0)	
Hospitalization stays, last 5 years, frequency (%)			
Yes	4 (26.7)	7 (46.7)	0.256
No	11 (73.3)	8 (53.3)	
Pulmonary function tests			
FVC (%), mean (SD)	61.5 (28.2)	65.8 (19.1)	0.632
FEV1 (%), mean (SD)	63.7 (28.0)	61.3 (19.2)	0.780
FEV1/FVC ratio (%), mean (SD)	108.9 (13.6)	105.7 (15.5)	0.544
Pulse oximetry			
SpO_2_ (%)	96.7 (3.0)	97.1 (2.1)	0.718

Abbreviations: CG: Control group; EG: Experimental group; SD: Standard deviation; %: Percentage; BMI: Body mass index; FVC: Forced vital capacity; FEV1: Forced expired volume at 1 s; SpO_2_: Oxygen saturation.

**Table 3 healthcare-10-01680-t003:** Effect of the treatment on respiratory parameters in the experimental and control group.

		Time Mean (SD)		Mean Difference (95%CI); Effect Size (*d*)	
		Pre	Post	Within-Group Differences	Between-Groups Differences (Post)
Pulmonary function tests					
FVC (%)	EG	65.8 (19.2)	77.7 (29.9)	11.9 **^b^**(−23.2 to −0.7)	11.8 (−8.9 to 32.5)
	CG	61.5 (28.2)	65.9 (25.2)	4.4 (−10.1 to 1.3)	
FEV1 (%)	EG	61.3 (19.2)	74.7 (27.8)	13.4 **^b^**(−30.6 to 3.8)	12.1(−9.5 to 31.6)
	CG	63.7 (28.0)	62.6 (27,00)	−1.1 (−5.2 to 5.5)	
FEV1/FVC ratio (%)	EG	105.7 (15.5)	99.5 (21.2)	−6.2 (−7.1 to 10.7)	−0.1(−11.1 to 17.6)
	CG	108.9 (13.6)	99.5 (21.2)	−9.4 (−1.4 to 18.0)	
Pulse oximetry					
SpO_2_ (%)	EG	97.1 (2.1)	97.9 (1.9)	0.8 (−1.3 to −0.3)	1.67 **^a,b^**(−0.5 to 1.5); *d* = 0.8
	CG	96.7 (3.0)	96.2 (2.3)	−0.5 (−0.2 to 1.3)	

Abbreviations: EG: Experimental group; CG: Control group; SD: Standard deviation; %: Percentage; CI: Confidence interval; FVC: Forced vital capacity; FEV1: Forced expired volume at 1 s; SpO_2_: Oxygen saturation; **^a^**: *p* < 0.04 corrected by Benjamini-Hochberg; Cohen’s d effect size was calculated for comparisons where statistically significant difference was obtained; Statistical power for comparisons of 80%; **^b^**: Meets criteria for minimal clinically important difference.

## Data Availability

The data that support the findings of this study are available from the corresponding author, [A.A-G.], upon reasonable request.

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
