# Peer review of "Clinical Changes of Respiratory Parameters in Institutionalized Older Adults after a Physiotherapy Program Combining Respiratory and Musculoskeletal Exercises"

_healthcare, 2022, doi:10.3390/healthcare10091680_

Round 1
Reviewer 1 Report
I would like to express my gratitude regarding the opportunity to review this manuscript.
It is an interesting study, but at this stage the manuscript requires significant improvements. Below suggestions with page indication (please consider insert the lines in v2):
1 - Please consider instructions for authors and journal template (title in uppercases). Also, the template header presents 2021, please correct.
1 - The affiliations numbers format in authors is not standardized. Please review.
1 - Please review signals separating keywords and in the end.
2 – Ref 9 before 8 – Please review.
2 – Please correct before “[13]”, also in page 3 before “[24]” and “[25]. Please carefully review all manuscript.
2 - The aim of the study should be clearly stated, and afterwards some hypothesis formulated.
2 – Please review, it seems more than one space before “men”.
2 - 2.4 – Any familiarization before intervention? What about conditions? Place, conditions (temperature, humidity). Time of day? Subjects’ routines and past experience? Nutrition? All that relates to intervention and data collection should be described in detail.
3 – “ADL” should be in full in the first appearance in the manuscript.
3 – Inclusion and exclusion criteria should be described in detail.
4 – Table 1 – In English is not “1º” but “1st”. Please correct in all cases.
5 – Figure 1 text font and size should be according to the journal template.
5 – In this page and in methods, all instruments’ procedures, and manufacturers information (with city and country) should be indicated.
6 – It Is suggested the written of the end paragraphs of “outcomes”.
6 – Please consider merging 2.6 and 2.7 and write the “statistical analysis” in detail.
7 – Please review figure 2 format and quality. For example, in subtitle “figure 2” repeated.
7-8 – Please review table 2 content and format.
9 – Table 3 – “a,b” and other examples. Readers should understand what represents (no reference in table).
10 – Please improve the quality of fig 3.
10-11 – Discussion section. Globally, the discussion section can be improved (for example “minimum age 60 and maximum 91” / “70% woman” / “Medication” / “Hospitalization” are topics to be addressed and some possible limitations). The deepening of the topic under study is very important. Also, some paragraphs with few text, contrary to others. Please consider standardization aiming providing readers better conditions for interpretations.
11 - Please consider developing the conclusions section with the main findings and clear and direct take-home messages.
12 – In “conflicts of interest” – Please remove “
12 – Appendix A should be described in the text, or with table or figure, for readers to relate to the rest of the text.
12 – References - Please review the instructions for authors and journal template and correct accordingly.
Please carefully review the English throughout the manuscript.
Reviewer 2 Report
Our questions are associated to experimental design and statistics:
Because both groups belong to the same place, explain what mesaures were taken to avoid internal validity threats associated
Sample size is admited to be a limitation, consequently, statistical options must be in accordance:
- Shapiro-Wilk, instead of Kolmogorov-Smirnov must be used
- non-parametrics are more adequate
- if there is a data distribution tendentially normal, two-sided probability error must be considered
Because mainly biological data are compared, effect size r, instead o Cohen's d, should be applied
Because hypertension and psychiatric medication can affect ANS, vagal tone and barorreceptors, samples need to be characterized in these variables
Experimental group has pre-test with higher SpO2 and FVC (the last one with smaller SD), which may have affected enhanced capacity of this group to benefit with specific program, this is an internal validity threat and must be considered a limitation of the study
Figure 3 has a very relative interest, instead, we strongly suggest boxplots or line graphs (with SD), for pre post, with both groups, per dependent variable
Considering all previous observations, in "Discussion", first paragraph is considered somewhat speculative
We strongly recommend authors to:
- remake all statistics, and review discussion and conclusion according
- include all limitations, particularly those related with groups common origin
Round 2
Reviewer 1 Report
Dear authors,
Thank you for considering my suggestions and incorporating them into the manuscript, which globally improved, congratulations.
Regarding the next version, suggestion is made to carefully review the manuscript and consider the journal template and instructions for authors. Also, it is important to look for the identification of small details to be improved and at the same time in the search for improvement of details in English, which overall is ok.
Below suggestions regarding details related to this new version. Lines are indicated.
58 – “severity[8].” – Please insert space before “[]”.
Page 4 – “physiotherapist provided” – It seems more than one space between words, please confirm.
130 – Please describe the professionals who implemented and supervised the intervention program and also the evaluations. Namely academic background and experience are important.
134 – Information is relevant regarding the subject’s daily routine: No exercise Tuesdays and Thursdays? Other activities? Nutrition remained the same in intervention days? No smoking, caffeine? No medicine that may influenced the program and evaluations? Did all subjects performed all intervention session or missed sessions? These and other pertinent information’s should be described.
177 – Text between table and figure introducing figure 1 is suggested.
183,189 and other lines – The lines of the figure do not seem with the same format, please review.
296 – Please insert space “ratio[29]”.
300 – Please review line spacing.
Page 10 – Figure 2 – Please standardize text. For example “=” spaces before and after. In text after figure no space (line 309).
Page 10 – Figure 2 – Please increase box size with “Not willing to participate” since this part of text is cut.
307 & 210 – Legend in upper and lowercase, please standardize considering the journal template and instructions for authors.
309-312 – The text should be justified.
313 – Please review line spacing between title and table.
364 – Please review the table content, namely positive and negative values.
365-369 – Please review legend format considering the journal template and instructions for authors.
393 – All the discussion section text should be justified and with text without spaces between paragraphs.
401-402 – Please consider rephrasing “and had done exercise regularly in the last years” and improved the English throughout the manuscript, despite the evidence that globally is ok.
427 – “strength[37],” – Please correct.
437 – “airflow[41].” – Please correct.
441 – “rest[13].” – Please correct.
490 – Please place “and” before last author in each contribution.
495 – End point is missing.
References - Please review the instructions for authors and journal template and correct accordingly. Some examples (details): Size of letter and paragraph; REF 4 year not in bold.
